# Unconscious Processing of Greenery in the Tourism Context: A Breaking Continuous Flash Suppression Experiment

**DOI:** 10.3390/ijerph20032144

**Published:** 2023-01-24

**Authors:** Xiang Huang, Hao Jiang, Ming Lv

**Affiliations:** 1School of Tourism Management, South China Normal University, Guangzhou 510006, China; 2South China Ecological Civilization Research Center, South China Normal University, Guangzhou 510006, China

**Keywords:** recovery, greenery, unconscious, tourism scene, breaking continuous flash suppression (b-CFS)

## Abstract

Both tourism and nature have been proven to contribute to people’s physical and mental health. Most studies have discussed their positive effects at the conscious level, but the unconscious mechanisms underlying these effects remain under-investigated, especially in the tourism context. Using a psychological experimental paradigm called breaking continuous flash suppression (b-CFS), this study tested how the proportion of greenery in an environment influences people’s perceptions of tourism sites and compared the effects of different proportions of greenery on participants’ unconscious responses to tourism sites. The results suggest that the presence of greenery improves the participants’ unconscious perceptions, and that this effect is due to greenery as an element of the natural world, rather than to green as a color. These findings enhance the understanding of the role that the unconscious response plays in the effect of nature on human health and may have managerial implications for the tourism industry.

## 1. Introduction

Job-related pressure has negative psychological and physical effects, and individuals with busy lives may experience negative emotions and even burnout. Tourism gives individuals an opportunity to recover their physical and mental health. People engaging in tourism seek rest and relaxation and to restore the vitality lost through daily living and work [1]. As such, the desire for recovery is an important motivation for engaging in tourism [2]. In summary, people engage in tourism activities to cope with and recover from the pressures of daily life.

A period of positive or negative rest neutralizes fatigue [3]. Recovery time or activities, which are essential for neutralizing fatigue, allow an individual to return to a base level of functioning [4]. Recovery from fatigue can occur during tourist trips [5]; such tourism is a key method of recuperating from the fatigue associated with work and daily life [6]. Some studies have examined the relationship between tourism (e.g., vacations) and recovery. According to the effort-recovery model [7], tourism activities are restorative in that they allow tourists to avoid the effort of their daily routines. Therefore, an important prerequisite for recovery is that tourists do not call on the functional systems used in their daily lives [7]. However, the conservation of resources theory [8] suggests that vacations allow individuals to replenish their valuable resources (e.g., energy) to achieve recovery, and that rest from stressful work prevents further the depletion of internal resources. Leisure and entertainment activities in nonworking hours can also help individuals build their internal resources. Although these two theories explain the mechanism of individual recovery, they ignore the role of the environment. However, in the field of environmental psychology, attention recovery theory (ART) [9] is a popular theory that explains the relationship between recovery and the environment [6]. It suggests that the mental fatigue caused by the constant use of directed attention in daily life can be recovered by engaging in activities that do not require much effort or energy. ART-based studies have shown that restorative activities can help people recover from stress and mental fatigue [10,11,12]. However, none of these theories are based on unconscious mechanisms.

Nature is known to aid the recovery of mental and physical health in humans [13]. Numerous studies have found that nature is beneficial to humans’ psychological, physical, and overall well-being [14]. Studies have also shown that environments with different levels and forms of natural elements provide many physical and psychological benefits and an enhanced sense of recovery [15,16,17,18,19]. Using ART, stress recovery theory, or biophilia theory as a framework [9,20,21], studies have manipulated research factors at the conscious level by asking participants to view natural or nonnatural images or films, and then the studies have measured their attention recovery [22] or their attention recovery, stress, and other physiological indicators [10,23]. Other studies have required participants to walk or engage in activities in natural or nonnatural environments and then measured their emotional experiences, stress and attention levels, pulse rate, diastolic blood pressure, salivary cortisol concentration, and other variables [11,24,25]. Most such studies have assessed the restorative effects of nature at the conscious level. Although Kaplan [26] observed that nature’s restorative influence can occur at the unconscious level—that is, people process natural information unconsciously—few studies have examined the effects of unconscious perception. For example, Lin et al. [27] explored the impact of the awareness of trees on recovery in urban environments and showed that those with a greater awareness of natural elements reported better recovery than those without this awareness.

Despite significant research on the impact of nature on recovery, the unconscious process underlying the cognition of nature remains inadequately understood, especially in the tourism context. This lack of understanding leads to questions such as: do we process natural stimuli at the unconscious level, and do we process it at different speeds than we process input at the conscious level? The answers to these questions may explain why studies have found that recovery performance is better in natural stimulus conditions than in nonnatural stimulus conditions. Specifically, an individual uses fewer cognitive resources, converts visual information into awareness more rapidly, and is better able to restore their cognitive resources when dealing with natural rather than nonnatural stimuli, and these benefits improve their subsequent performance of cognitive appraisal tasks.

Given that both tourism and nature contribute to health recovery, our research focuses on greenery (mainly grasslands and trees) in a tourism context. Greenery is a common and important natural element in tourism locations and, more broadly, it is a representative feature of people’s concepts of nature. Moreover, many studies based on ART have found that most of the places that are conducive to people’s recovery are green settings with plants, such as forests, wilderness environments, and parks [26,28,29,30,31]. This paper explores the following question: Do human beings perceive natural stimuli at an unconscious level? In other words, will adding natural elements (trees or grass) increase the appreciation of tourism scenes?

This paper’s main contribution is that it fills the knowledge gap regarding the unconscious processing of natural elements in the tourism context. Moreover, it helps explain the common-premise hypotheses of ART, stress recovery theory, biophilia theory, and other theories on the positive role of nature in human evolution.

## 2. Tourism, Vacations, and Recovery

Tourism is a nonmedical intervention that can improve physical and mental health; accordingly, tourism research has increasingly focused on the benefits of tourism. Such research has suggested a theoretical basis for the benefits of tourism. According to the effort-recovery model, individuals require rest to recover to a baseline level after daily physical and mental effort, and vacations can promote such recovery and prevent serious harm to physical and mental health [7]. The conservation of resources theory divides individuals’ resources into categories, namely, material, conditional, individual characteristics, and energy, and emphasizes vacations as a means of recovering these resources [8].

Some researchers have used a pre- and post-test design to measure perceived health and mental well-being before and after a vacation [32,33,34,35]. Studies have shown that vacations can positively affect personal cognition, creativity, and subjective well-being [36,37,38]. Some studies have also found that even short weekend trips can promote recovery; however, longer trips provide more opportunities for recovery [32,34,39,40], with the effects lasting for days or weeks [32,34,39]. Although the degree of recovery varies by destination, many studies have shown that natural environments yield better recovery outcomes than artificial environments [41,42,43]. In most of these studies, the researchers used images and videos of natural environments or natural elements as the objects and measured the effect of nature on recovery using self-reporting and psychological and physiological indicators. However, the unconscious mechanism underlying the positive effect of nature remains unexplored.

## 3. Materials and Methods

### 3.1. Breaking Continuous Flash Suppression: Measuring Unconscious Processing

Jiang, Costello, and He [44] developed breaking continuous flash suppression (b-CFS) as an experimental method for measuring unconscious processing. In this technique, a series of images is continuously and rapidly flashed in front of the participant’s dominant eye, while a target image with increasing contrast is presented in the corresponding position in front of the nondominant eye. The participant is then asked to press a button immediately when they observe any part of the target image. As an interocular inhibition paradigm, this method allows participants to see different stimuli with each eye through a stereoscope. Because of the technique’s highly dynamic nature and the strong contrast of the suppression images, participants have reported that they could not see the target stimulus at the beginning of the experiment. However, the suppressed target stimulus is still processed to a certain extent at the unconscious level. Different target images have been shown to break through the interocular inhibition at different times [45]. In b-CFS, the ability of a stimulus to break through is assumed to indicate the degree of unconscious processing. Studies have shown that evolutionarily significant stimuli, such as frightening faces containing threat information and environments or objects beneficial to survival and reproduction, break through faster than other types of stimuli [46]. The b-CFS paradigm was used to test whether the time to break through when viewing a tourism scene (stimulus) was affected by the level of greenery in the scene. The time when the target image broke through was recorded as the participant’s reaction time.

### 3.2. Stimuli

Ethics committee approval (SCNU-STM-2022-001) was obtained from the Institutional Ethics Committee of South China Normal University before this study commenced. This study used a single-factor within-subject design. The stimulus design was based on the work of Xiao, Wei, and Li [47], who proposed a visible green index divided into five sections to illustrate different quantities of greenery: quite poor (the visible green index is less than 5%); poor (5–15%); some (15–25%); more (25–35%); and very good (more than 35%). In this experiment, the independent variable was the proportion of greenery in the tourism scene. We used four levels of greenery, following the method of Xiao, Wei, and Li [47], who found 35% to be the limit of the human perception of greenery. The four levels corresponded to four groups of images: Group 1, which is the no-greenery group, viewed original images with no greenery (green index ~0); Group 2, which is the low-greenery group, viewed images with a low level of greenery (<35%); Group 3, which is the high-greenery group, viewed images with a high level of greenery (>35%); and Group 4, which is the control group, viewed images in which the greenery in the images used in the low-greenery group were replaced with a green block (Figure 1). The dependent variable was the time it took to break through interocular inhibition, which is the participant’s reaction time.

We selected 20 images from copyright-free websites and used Photoshop 2021 to add grass cover, grass and tree cover, or a green block to the image, resulting in 80 images distributed among the four groups as the stimulus. We ensured that the selected images contained only buildings that were tourist attractions (i.e., they contained no people, other man-made objects, or natural elements) and depicted sunny weather.

Because emotional information can affect unconscious processing [48,49,50], 40 participants were recruited to evaluate the arousal and pleasure associated with the selected images before the formal experiment commenced. In this assessment, the participants were asked to rate the images, which appeared on the screen with the Self-Assessment Manikin [51,52] (Figure 2). The participants were asked to rate pleasantness and excitement according to their intuitive experience with the images, with 1 being the least pleasant/exciting, 5 being neutral, and 9 being the most pleasant/exciting. After processing the arousal and pleasure scores for each image, a one-way ANOVA was conducted, and 12 groups of images (48 images) with no significant differences in the degrees of arousal (F (3, 47) = 1.497, *p* = 0.229) and pleasure (F (3, 47) = 0.869, *p* = 0.465) were selected (Table 1).

## 4. Results

### 4.1. Experiment 1

#### 4.1.1. Experimental Instruments and Materials

Following the methods of Anderson et al. [50], the matching color tool in Photoshop 2021 was used to adjust the contrast of the 48 tourism images selected in the pre-experiment for consistency. Next, MATLAB R2020a was used to generate a set of 10 graduated contrast images from each of the 48 initial images, with contrast levels of 10%, 20%, 30%, 40%, 50%, 60%, 70%, 80%, 90%, and 100% (original image). These pictures were used as the stimulus images. In addition, 10 black-and-white Mondrian noise figures were generated using MATLAB R2020a.

The experimental process was adapted from Shang et al. [53] and programmed and controlled using MATLAB R2020a. A monitor with a 27-inch display, 60 Hz refresh rate, and 1920 pixel × 1080 pixels was used to display the instructions and stimuli. A stereoscope was used to map the images displayed on the screen to the participants’ left and right eyes. During the experimental procedure, the display background was gray (RGB: 128, 128, 128), and the central fixation cross (+) was black (0.75° × 0.66°). The stimulus and noise images were surrounded by two frames (21.18° × 13.47°). During the experiment, the images were displayed within the frames.

#### 4.1.2. Participants and Procedure

We recruited 35 college students (21 female, 20–24 years old, M = 21.60 years old, SD = 1.33) to participate in Experiment 1. All of the participants had normal or corrected-to-normal visual acuity, without color blindness or color weakness. They had not participated in similar experiments and were compensated after completing this experiment. The experiment and its possible effects were clearly described to them, and the participants provided consent before the experiment began.

The whole experiment took place in a quiet and dark room. Before the experiment, the Dolman method [54] was used to determine the participants dominant eyes. Each participant sat in front of the monitor, rested their head on a chin rest, and adjusted their position so that their eyes were about 60 cm away from the screen, as shown in Figure 3. The experimental program began automatically 2 s after the participant was ready, as shown in Figure 4. The Mondrian images (17.35° × 10.62°) were displayed at a frequency of 10 Hz at the position corresponding to the participant’s dominant eye, and a tourism scene image (11.77° × 8.88°) with gradually increasing contrast was randomly presented above or below the fixation cross corresponding to the nondominant eye. The contrast level increased linearly from 0% to 100% within 1 s and then remained constant until the participant responded. The participant was asked to press the “Z” key as quickly and accurately as possible if they observed any part of the tourism scene and then to judge the position of the image. If it was above the fixation cross, they were to press the up-arrow key; if below, they were to press the down-arrow key. The time until the “Z” button was pressed after the first image presentation was recorded as the reaction time, and a recording was made of both correct and incorrect position judgments. In the formal experiment, 96 images were presented: 48 images were used, and each image was displayed twice. Fifteen practice tests were carried out before the formal experiment. Referring to the work of Shang et al. [53], the images were presented in a pseudo-random order, according to a randomly generated sequence. Each subject was shown the same fixed sequence of images.

#### 4.1.3. Results

We excluded the reaction times associated with misjudgments of position and those more than three standard deviations above or below the mean reaction time for all of the participants, or the mean reaction time in each group. The average reaction time of each participant in the four groups was then calculated.

Repeated measures ANOVA were used to investigate the participants’ reaction times by level of greenery coverage in the images. As shown in Figure 5, the greenery level had a significant effect, F (3, 102) = 25.614, *p* ≤ 0.001, ηp2 = 0.430. Further pairwise comparison (Bonferroni corrected) showed that the reaction time for the no-greenery group images (M = 1.431, SD = 0.051) was significantly greater than that of the other groups (low-greenery group: M = 1.241, SD = 0.042; high-greenery group: M = 1.236, SD = 0.041; control group: M = 1.324, SD = 0.048; *p* < 0.05 for all). The reaction time for the low greenery group images was significantly less than that for the control group images (*p* ≤ 0.001), whereas the reaction times did not differ significantly between the high-greenery and low-greenery groups. The reaction time for the high-greenery group was significantly shorter than that for control group (*p* ≤ 0.001).

In Experiment 1, black-and-white suppression images were used. The participants’ reaction times to both the low-greenery group and high-greenery group images, which contained increasing levels of greenery, were significantly shorter than their reaction times to the images in no-greenery group and control group, which is consistent with the finding by Lin et al. [27] that different greenery levels affect unconsciousness processing. However, although the reaction times for the low-greenery group and high-greenery group differed significantly from that for the control group, the reaction times for Group 4 significantly differed from those for the no-greenery group. Therefore, we could not deduce whether the color green, the physical characteristics of the greenery, or the natural aspects of the greenery were the reasons for the differences in the reaction times.

In Experiment 1, the tourism scenes were colorful, whereas the suppression images were black and white. Therefore, the effect of greenery on reaction time may have depended on the perception of color. Hong and Blake [51] found that color suppression images inhibit the processing of color information in the target stimuli more strongly than black-and-white suppression images. To explore whether the level of greenery coverage still affected reaction times when the stimulus color information was more strongly suppressed, we conducted Experiment 2, in which we used colorful suppression images to further explore the unconscious processing of a tourism scene with different levels of greenery.

### 4.2. Experiment 2

#### 4.2.1. Experimental Instruments and Materials

The experimental steps were the same as in Experiment 1 except, that MATLAB 2020 was used to generate 10 colorful (rather than black-and-white) Mondrian suppression images.

#### 4.2.2. Participants and Procedure

Another 35 participants were recruited, and all were given a clear explanation of the experiment and consent requirement. One subject dropped out due to discomfort with the experimental process, so the final valid dataset is based on the reaction times of 34 participants (20 female students, 20–24 years old, M = 21.65 years old, SD = 1.29). The selection criteria and remaining steps were the same as in Experiment 1.

#### 4.2.3. Results

As in Experiment 1, the reaction times associated with positional errors and those three standard deviations, above or below the mean reaction time for all of the participants, or the mean reaction time for each group, were removed. The average reaction time of each participant in the four groups was then calculated.

Repeated measures of ANOVA was used to investigate and compare the participants’ reaction times according to the level of greenery coverage. As shown in Figure 6, the greenery level had a significant main effect, F (3, 99) = 10.250, *p* ≤ 0.001, ηp2 = 0.237. A further pairwise comparison, with Bonferroni correction, showed that the reaction time to the images in the no-greenery group (M = 1.883, SD = 0.091) was significantly greater than that to the images in the high-greenery group (M = 1.755, SD = 0.077; *p* = 0.035), but not significantly different from the reaction times to the images in the low-greenery group (M = 1.756, SD = 0.072) or the control group (M = 1.910, SD = 0.082). The reaction time to the images in the low-greenery group was significantly shorter than that to the images in the control group (*p* ≤ 0.001). There was no significant difference between the reaction times to the images in the low-greenery group and the high-greenery group. The reaction time to the images in the high-greenery group was significantly shorter than that to the images in the control group (*p* = 0.033).

The results of Experiment 2 showed no significant difference in the reaction times to the images in the control group and the no-greenery group or between the groups with high (high-greenery group) and low (low-greenery group) levels of natural greenery. The only significant differences were between the high-greenery group and the no-greenery group and the control group. These results indicate that the shorter reaction time to the images with natural greenery in Experiment 1 were not merely a reaction to the color green, but also to the natural aspect of the greenery. We attribute the lack of a significant difference between the reaction times to the images in the no-greenery group and the low-greenery group in Experiment 2 to the appearance of the greenery in the latter group, which lacked the contour of grass and appeared similar to flat ground. However, the lack of a significant difference between the high-greenery group and the low-greenery group in Experiment 2 may have been due to the added greenery—trees—in the high-greenery group images that might have blocked some of the features of the tourist attraction, increasing reaction times.

Unlike the results of Experiment 1, the results of Experiment 2 indicated that the influence of greenery can be attributed not only to color, but also to it being a natural element. However, the difference in the reaction times for high and low levels of greenery coverage was not significant, and greenery that occluded the tourist attractions may have affected the participants’ perceptions.

## 5. Discussion

### 5.1. Unconscious Processing Effects of Greenery

The b-CFS paradigm was used to show that the reaction times to the tourism images with low and high levels of greenery coverage (low-greenery group and high-greenery group, respectively) were shorter than the reaction times to the images without greenery coverage (no-greenery group) and to the control images with a green coating (control group) when black-and-white suppression images were used, but there was no difference in the reaction times between the images with different levels of greenery coverage. When colorful suppression images were used, the reaction times to the images in the high-greenery group were significantly shorter than those to the images in the no-greenery group and the control group, whereas no significant differences were observed between the reaction times to the images in the low-greenery group and those in the other groups. In addition, the reaction times in the experiment with the colorful suppression images were significantly longer than those with the black-and-white suppression images, indicating that the inclusion of color in the suppression images enhanced their suppressive effect.

The results of Experiment 1 showed that the presence of greenery affected the participants’ unconscious processing of tourism scenes. The differences in their reaction times to tourism scenes with different levels of greenery coverage suggest that the participants were engaging in unconscious processing. Studies have found that the unconscious perception of urban greenery can affect how individuals evaluate perceived restoration [27]. Furthermore, the pre-experimental screening of the target images eliminated the influence of irrelevant variables, such as the degree of arousal and pleasure, thus confirming that the level of greenery affects the recognition of tourism scenes without any effect related to color variations or other factors in the images.

In contrast to Experiment 1, Experiment 2 used colorful suppression images. The results indicate that the participants’ unconscious processing was affected not only by the color green, but also by the presence of greenery as a natural element. The reaction times in Experiment 1 were significantly shorter than those in Experiment 2, possibly because of the influence of color (green) as a significant characteristic of greenery. However, Experiment 2 produced the effect, even when color information was suppressed, thus strongly demonstrating that natural elements (i.e., greenery) have an impact at the unconscious level. However, the reaction times to the images in the low-greenery group did not differ significantly from the reaction times to the images in the other groups in Experiment 2. We attribute the lack of a significant difference between the reaction times to the images in the no-greenery group and the low-greenery group to the flatness of the green coverage added to the low-greenery group images. The lack of contour may have made it difficult to distinguish the grass from plain ground, thus reducing the perception of a difference. The differences between the low-greenery group and high-greenery group in both experiments might be due to the canopy of trees added to the high-greenery group images; specifically, the trees might have obstructed the tourist attraction and thus reduced the participants’ processing speed.

Thus, we conclude that greenery, as a representative natural element, promotes an individual’s cognitive processing of tourism scenes. The participants in our study were more likely to perceive the tourism scenes because of the enhancing effect of greenery on their unconscious processing.

In recent years, considerable research has explored the benefits of nature, using ART, stress recovery theory, or biophilia theory as their theoretical foundations. Although the physiological and psychological benefits of nature are attractive research topics, most studies have examined the effects at the conscious rather than the unconscious level. Solid evidence of an evolutionary basis for the human preference for nature is lacking [9,21,22]. By exploiting the benefits of unconscious processing in a research setting, this study shows that people process natural elements at an unconscious level. In other words, humans have an unconscious advantage in processing the natural elements of tourism scenes, and this might be rooted in evolution, thus lending support to theories on the benefits of natural elements in tourism. Increasing the natural elements (e.g., greenery) in a tourism setting may allow visitors to recover via mechanisms at the unconscious level. Accordingly, information about a tourism location can quickly enter a tourist’s consciousness, so the tourist becomes aware of their surroundings with less conscious effort, allowing them to replenish their directed attention, which is frequently used in daily life [20].

### 5.2. Limitations and Recommendations for Future Research

This study has some limitations. First, only 12 of the 20 initially selected images met the requirements. Therefore, each image was repeated twice in the formal experiments, so the participants may have become familiar with the images. This familiarity might have affected the results. Second, the difference between the images in the low-greenery group and the high-greenery group (low and high greenery coverage) was slight, and the greenery in the images in the high-greenery group partially obscured the building, which may have affected the results. This may explain why the reaction times to the images in these groups did not differ significantly in Experiment 2. Additionally, the treatment of the control images in the control group may not have been sufficiently natural, and the differences between the image and a natural scene might have affected the participants’ reaction times. Moreover, this study did not consider the influence of the participants’ backgrounds and willingness to participate in environmental or tourism activities on the experimental results. According to some studies [55,56,57,58] and our understanding of this field, a person’s background (e.g., being from a rural area, where natural elements are abundant, vs. being from an urban area, where natural elements are less abundant) may affect their perception of natural elements. Future research could focus on material processing (e.g., place the trees in the foreground to avoid obscuring the buildings), selection (e.g., different types of buildings with different greenery), and more detailed descriptions and comparisons of participants’ demographics to improve the accuracy of the experimental results. Finally, even if our conclusion is true in tourism scenarios, we cannot conclude that it is true in other scenarios or in response to different elements of nature. This research only shows the impact of greenery in the tourism context. Future research could explore different scenarios and the effects of different natural elements to identify their effects on unconscious processing during interactions with nature.

## 6. Conclusions

This study examines whether natural elements (e.g., greenery) in a tourism context are processed at the unconscious level. Interestingly, the results show that greenery as a natural element, rather than the color green, influences unconscious processing. Therefore, recovery in a natural environment may not require conscious involvement, which confirms Kaplan’s view.

The effect of nature on unconscious processes also has important practical implications. The results suggest that humans can recover their health in nature. In practice, this may offer a new approach for tourism operators: properly increasing the level of greenery (i.e., not occluding tourist attractions) in tourism settings can enhance visitors’ unconscious awareness and thus facilitate their recovery. However, further exploration is needed, as the effects of nature may vary by the type of natural element.

## Figures and Tables

**Figure 1 ijerph-20-02144-f001:**
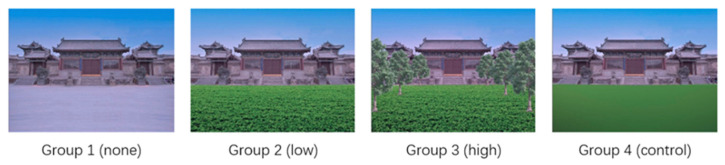
Four levels of greenery in a tourism scene.

**Figure 2 ijerph-20-02144-f002:**
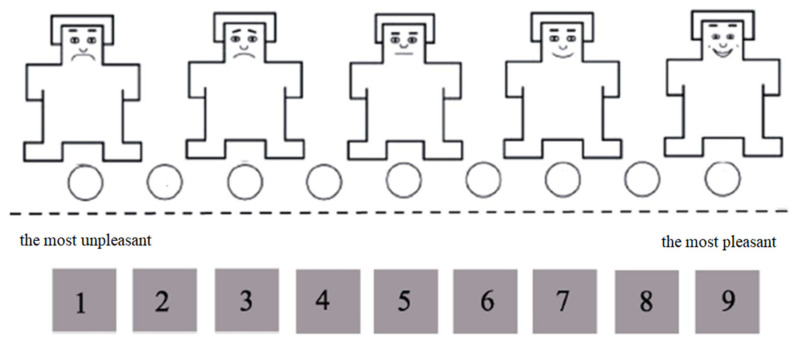
Pleasure and arousal evaluations.

**Figure 3 ijerph-20-02144-f003:**
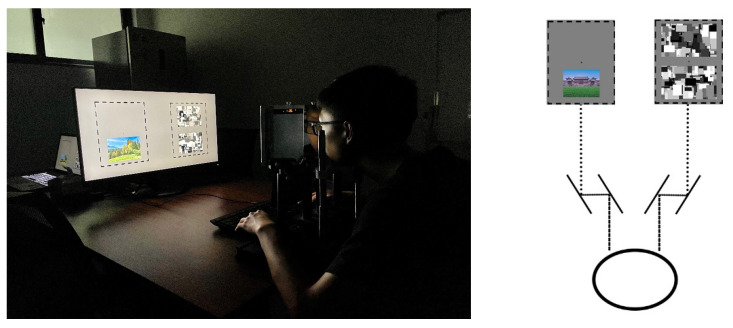
Illustration of the stereoscope setup.

**Figure 4 ijerph-20-02144-f004:**
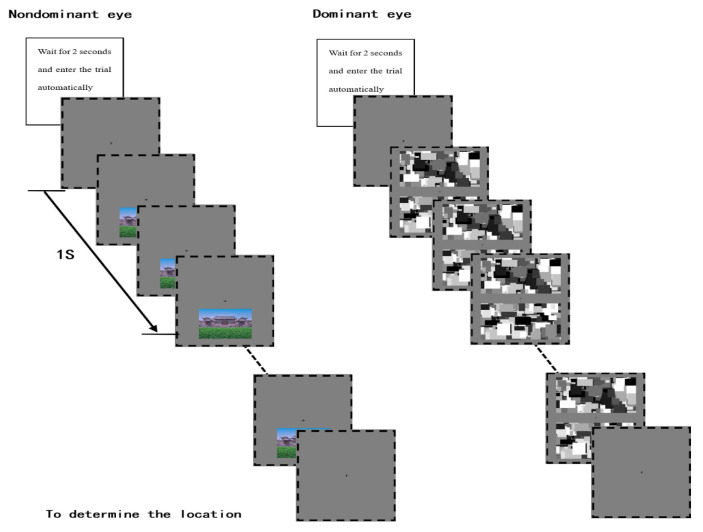
Schematic representation of the experiment.

**Figure 5 ijerph-20-02144-f005:**
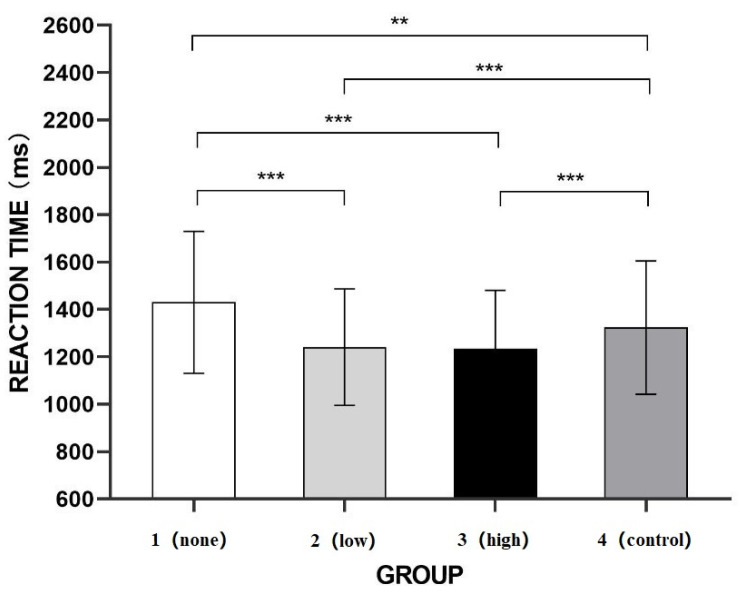
Reaction times to tourism scenes with different levels of greenery and black-and-white suppression images. *** *p* < 0.001 **, *p* < 0.01.

**Figure 6 ijerph-20-02144-f006:**
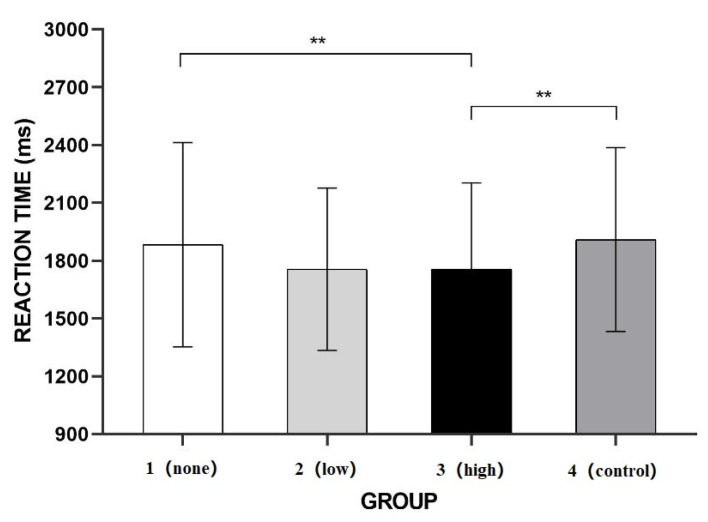
Reaction times to tourism scenes with different levels of greenery and colorful suppression images. ** *p* < 0.01.

**Table 1 ijerph-20-02144-t001:** Appraisal of the arousal and pleasure evoked by the image stimuli.

Coverage of Greenery	Arousal		Pleasure	
	M	SD	M	SD
None	5.08	0.793	6.00	0.853
Low	5.58	0.669	6.42	0.996
High	5.33	0.779	6.08	1.311
Control	5.00	0.739	5.75	0.866

## Data Availability

The data are not publicly available due to privacy protection for essential workers.

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
