# Peer review of "Unconscious Processing of Greenery in the Tourism Context: A Breaking Continuous Flash Suppression Experiment"

_ijerph, 2023, doi:10.3390/ijerph20032144_

Round 1
Reviewer 1 Report
Page 1 lines 34 to 42: The authors expand on the existing literature on tourism and recovery by discussing three major theories: effort-recovery, resources, and attention recovery. However, the transition between the three is confusing. It is also unclear how different theories relate to this study.
Page 2 lines 50 to 74: It would be helpful if the authors could provide a more organized literature review on the impact of nature on recovery at both the conscious and unconscious levels. The current flow is difficult to follow. For example, reference [20] is about the influence at the unconscious level, which follows directly by a discussion about the influence at the conscious level without proper transition. This paragraph should be better streamlined.
Page 2 lines 75 to 80: Because greenery is the focus of this study, I believe there should be more background information about the relationship between greenery and health recovery. The context is currently thin.
Page 4 lines 156: More backgrounds regarding the arousal and pleasure of each image would be beneficial (how they are measured, why important, etc.).
Page 6 lines 215 to 224: It is preferable to specify the level of perceived greenery that each group corresponds to here. Readers may struggle to connect the experimental results to the groups described on page 4.
Reviewer 2 Report
The article proposes a topic that is relevant, and its approach is interesting. The paper as currently written may have these issues as follows;
Methodology / Experimental Design
The methodology appears to be sound, but the experimental setup may have potential issues. A key finding is the differences to reactions times comparing Group 1 pictures (higher time), to Groups 2 and 3 pictures (lower times) in Experiment 1. A question that comes to mind is if the sequence in which the experiment was performed has a learning curve effect. Is it possible that the respondents simply take a longer time to react to Group 1 pictures because they are “learning” how to act in the experiment, and that their reaction times improve when viewing Groups 2 and 3 pictures? (This assumes that the experiment was performed in the sequence Group 1, Group 2, Group 3, and Group 4). Another question that comes up is if the higher reaction times for Group 4 can then be due to a fatigue effect? This learning curve and fatigue effects combined could account for the U-shaped reaction times shown in Figure 5. It was mentioned that fifteen practical tests were performed before the experiment started, but it is not clear how these practical tests could mitigate these effects.
An alternative experimental design is to perform the test in this order: Group 3, Group 2, Group 4, Group 1 (with rest periods in between to prevent fatigue). The expected results will be increasing reaction times where Group 3 should show the lowest, and Group 1 should show the highest times to react.
Further, with a second and independent set of responders, the experiment can be repeated in the order of Group 1, Group 4, Group 2, Group 3 (with rest periods in between). The expected results will be decreasing reaction times where Group 1 is the highest, and Group 3 is the lowest times to react. (The Mondrian images can be from those as used in Experiment 2).
This may be able to provide results that show if there are any significance from the learning curve or fatigue effects.
Repeating the test in reversed order can also independently test if Group 1 or Group 4 will elicit the “higher” or “lower” reaction times when compared with one another.
Control
Limited information is provided on the participants demographics. At least two factors may affect the results; (1) The gender of the participants, (2) the background of the participants.
Gender issues may play a role because males and females may process visual cues differently, or have different reaction times to press buttons. It may be useful to analyze the results based on a gender split to show if these factor have or do not have such effects.
The background of the participants may be a factor if respondents have rural, suburban or urban backgrounds. Presumably, people who grew up in a rural environment surrounded by greenery may react differently to visual cues (either faster or slower) from those who grew up in an urban environment with less greenery. A review of the data along these lines may be useful to check if such an effect is present or not.
These gender or background factors can be checked in both in the original pre-experiment group of 40, and the subsequent experimental groups of 35 respondents.
It is understood that college students are used as a convenience sample. Perhaps one way to control this will be to survey these college students on how much they will be willing to spend on tourism activities over the next years. In this way, the practical effect of providing greenery around tourist areas can be better assessed. For instance, if students who are more willing to spend on tourism also shows faster reaction times to greenery, this shows that it is worthwhile for tourist operators to invest in greenery.
Others
I also wonder if there is an incongruity effect in the choice of pictures that may affect the experiment. For instance, people viewing pictures of an ancient building may not expect to view greenery, so they may react faster due to the unexpectedness of the sight. In contrast, people viewing pictures of a tourist resort fully expect to see greenery, so they may not be as simulated to react. Showing pictures of different tourist areas with or without greenery may answer this question.
Last, for the Group 3 picture, if the trees are placed in the foreground, that will avoid obscuring the buildings and remove the concern raised in the text.
Language
The standard of English used in the paper is excellent.
Reviewer 3 Report
The study is well written and have key implications. To make the study more understandable, I share my minor concerns, which should be incorporated before the paper is accepted.
Minor comments:
1. The study needs to provide contribution points in introduction section and logics. The author claims two contributions, however, the second one is based on the findings of first.
2. The author needs to indicate the theoritical importance of the study, in introduction section.
3. As the first contribution is “About the extent to which natural elements are processes ……”. How the findings are compared with this and highlight the term “at what extent” in the results.
4. The policy recommendation is well documented.
Round 2
Reviewer 1 Report
I do not have further questions about the manuscript.